# Potential Antagonistic Bacteria against *Verticillium dahliae* Isolated from Artificially Infested Nursery

**DOI:** 10.3390/cells10123588

**Published:** 2021-12-20

**Authors:** Xiaofeng Su, Siyuan Wu, Lu Liu, Guoqing Lu, Haiyang Liu, Xi Jin, Yi Wang, Huiming Guo, Chen Wang, Hongmei Cheng

**Affiliations:** 1Biotechnology Research Institute, Chinese Academy of Agricultural Sciences, Beijing 100081, China; wusiyuanjh@zjnu.edu.cn (S.W.); luliu_bio@163.com (L.L.); luguoqing007@163.com (G.L.); guohuiming@caas.cn (H.G.); 2Institute of Plant Protection, Xinjiang Academy of Agricultural Sciences, Urumqi 830091, China; liuhaiyang001@163.com; 3Hebei Technology Innovation Center for Green Management of Soil-Borne Diseases, Baoding University, Baoding 071000, China; jinxi2007@126.com; 4Qiannan Branch of Guizhou Provincial Tobacco Corporation, Qiannan 550100, China; wangyi327@126.com

**Keywords:** extracellular enzyme, nonribosomal peptides, cotton, resistant gene expression, biocontrol fungicides

## Abstract

As an ecofriendly biocontrol agent, antagonistic bacteria are a crucial class of highly efficient fungicides in the field against *Verticillium dahliae*, the most virulent pathogen for cotton and other crops. Toward identifying urgently needed bacterial candidates, we screened bacteria isolated from the cotton rhizosphere soil for antagonisitic activity against *V. dahliae* in an artificially infested nursery. In preliminary tests of antagonistic candidates to characterize the mechanism of action of on culture medium, 88 strains that mainly belonged to *Bacillus* strongly inhibited the colony diameter of *V. dahliae*, with inhibiting efficacy up to 50% in 9 strains. Among the most-effective bacterial strains, *Bacillus* sp. ABLF-18, and ABLF-50 and *Paenibacillus* sp. ABLF-90 significantly reduced the disease index and fungal biomass of cotton to 40–70% that of the control. In further tests to elucidate the biocontrol mechanism (s), the strains secreted extracellular enzymes cellulase, glucanase, and protease, which can degrade the mycelium, and antimicrobial lipopeptides such as surfactin and iturin homologues. The expression of *PAL*, *MAPK* and *PR10*, genes related to disease resistance, was also elicited in cotton plants. Our results clearly show that three candidate bacterial strains can enhance cotton defense responses against *V. dahliae*; the secretion of fungal cell-wall-degrading enzymes, synthesis of nonribosomal antimicrobial peptides and induction of systemic resistance shows that the strains have great potential as biocontrol fungicides.

## 1. Introduction

*Verticillium dahliae*, a soil-borne phytopathogenic fungus, is the causal agent of a destructive vascular wilt known as Verticillium wilt. This fungus was firstly reported in America and rapidly spread throughout the world [1]. It has a comparatively broad host range and can infect at least 200 dicotyledonous plant species, including cereal crops, trees and ornamental plants [2]. Among them, cotton, an important oil and fiber crop worldwide, were severely harmed and induced Verticillium wilt, which is known as “cotton cancer” and causes the serious increasing of cotton yield and quality [3,4]. Microsclerotia, the main survival structure of *V. dahliae* in fields, are very difficult to eliminate from soils and responsible for great economical losses annually [5,6]. Their high tolerance to extreme temperature and pH conditions enables them to survive for nearly 20 years [7,8]. Furthermore, the germination ratio is hardly influenced, although the microsclerotia is in a state of germination and dormancy up to nine times [9].

The disease cycle starts with germination of the microsclerotia, triggered by chemical signals from plant roots. After germination, hyphae develop, massively proliferate and envelop the surface of the roots, and some hyphae differentiate to form a special structure, the hyphopodium, which adhere to the root surface and subsequently penetrate the rhizodermis. Once inside the root, hyphae grow rapidly in the elongation zone and eventually colonize and clog the vascular system, resulting in the typical wilt symptoms, such as stunting, wilting, discoloration and defoliation. The fungus produce considerable microsclerotia, which are carried with plant debris into the field [1,10,11]. The microsclerotia, spread via contaminated seeds, plant residues and soil, could be accumulated in the soil and are overwhelmingly difficult to eliminate [12].

In general, the useful methods applied to prevent and control this fungi are crop rotation [13] and resistant cultivar [14]. Unfortunately, these methods are stringently confined in practice due to limited land and water resources. For instance, it is nearly impossible to plant rice in drought-prone areas [15]. The breeding of resistant cultivar will require a very long period of time and many resources [16]. In addition, another physiologic race of *V. dahliae* would dominate due to the large-scale single resistant variety for the long-term cultivation [2,17,18]. There are only a few available chemical fungicides that can combat *V. dahliae* induced wilting disease, many of which are detrimental to human health and the environment [19,20]. Antagonistic bacteria for biocontrol are a cost-effective approach and extensively used to improve crop yields [21,22,23,24]. They could inhibit the germination of microsclerotia and spores, promote growth of the plants and induce systemic resistance [25,26,27]. The most effective agents are usually isolated from the soils of plants or the plants themselves.

Two *Pseudomonas fluorescens* strains and one *Bacillus* strain, isolated from the cotton rhizosphere, greatly reduced the disease index and promoted the root length and dry weight of the cotton plant [28]. The plant-growth-promoting rhizobacteria, K-165 and 5-127 (*Bacillus* sp.) are efficient in reducing disease severity and significantly increased the height of treated potato and eggplants [29]. Four *Pseudomonas* spp. strains from the cotton and weed rhizosphere that were antagonistic against *V*. *dahliae* in vitro also obviously increased the growth variables of cotton and the seed cotton production in the field trials [30]. The disease incidence rates of *Verticillium* wilt of cotton were significantly suppressed upon treatment of *Bacillus subtilis* strain HJ5 [31]. The spore and microsclerotia germination and mycelial growth of *V*. *dahliae* are restricted by *Bacillus cereus* YUPP-10, which could significantly reduce the disease index of cotton in the pot experiment [32]. Four endophytic bacteria screened from tomato plants and their bio-control activity enhance seed germination and growth parameters of tomato and reduce the wilt disease in seedlings [33].

Although the global demand for ecoenvironment-friendly, sustainable agricultural methods has led to the increasing development of efficacious biocontrol agents, research on biocontrol agents to prevent and control of *Verticillium* wilt started relatively late in China, and the existing biocontrol resource pool is very limited. Moreover, some strains often mutate and lose efficacy during the practical application process. Thus, it is imperative to comprehensively excavate more efficient antagonistic bacteria and investigate their antagonistic mechanisms in depth. Toward this goal, here we (1) screened and characterized the potential antagonistic bacteria from artificially infested nursery against *V. dahliae* in-vitro tests; (2) evaluated the best candidate bacteria for antagonistic efficacy using pot experiment; (3) assayed activity of fungal cell-wall-degrading extracellular enzymes; (4) explored production of nonribosomal peptides; (5) investigated whether the defense response genes *PAL*, *MAPK* and *PR10* induced in cotton by antagonistic bacteria.

## 2. Materials and Methods

### 2.1. Soil Samples Collection

Soil samples were collected from the cotton rhizosphere in the artificially infested nursery (Langfang, Hebei province, China, 39°51′ N, 116°60′ E) set up to test cultivars for *Verticillium* wilt resistance by planting a field with cotton for a few years. Each year, the plant residue and solid culture medium were scattered on the field, then the field was uniformly plowed. The disease index for susceptible cotton variety Coker 312 grown in the field is up to 40–60.

### 2.2. V. dahliae Strain

The highly virulent *V. dahliae* V991 with a defoliating pathotype was obtained from Guiliang Jian at the Institute of Plant Protection, Chinese Academy of Agricultural Sciences (CAAS). This fungus was kept at −80 °C in 25% glycerol and recultured in CM liquid medium (casein acid hydrolysate 6 g/L, yeast extract 6 g/L, and sucrose 10 g/L) for 3 days.

### 2.3. Isolation of Potential Antagonistic Bacteria

The bacteria were isolated from soil samples using previously described methods [34] with minor modifications. Cooled, apportioned and mashed soil samples were serially diluted (10^−1^, 10^−2^, 10^−3^, 10^−4^, 10^−5^ and 10^−6^) with sterile distilled H_2_O, then 0.5 mL of liquid supernatant was spread onto potato dextrose agar (PDA) and incubated at 28 °C for 2 days.

### 2.4. In Vitro Antagonism Test

*V. dahliae* spores were harvested with a sifter (40 μm) (MILLEX^®^GP, Millipore, MA, USA), and the concentration was adjusted to 10^6^/mL. 10 μL spore suspension liquid and the candidate antagonistic bacteria were added on the PDA surface with 1 cm interval. Then, the plates were incubated at 28 °C for 2 weeks. Bacteria with strong antagonistic activity against *V. dahliae* were selected for the further analysis.

### 2.5. Molecular Identification of Antagonistic Bacteria and Phylogenetic Correlation

Genomic DNA of antagonistic bacteria was extracted and used for a polymerase chain reaction (PCR) [35]. The 16S rRNA sequences were amplified by the bacterial universal primers 27F (5′-AGAGTTTGATCCTGGCTCAG-3′) and 1492R (5′-GGTTACCTTGTTACGACTT-3′) [36]. The PCR reaction mixture included 1 μL of genomic DNA, 25 μL of 2× Taq PCR Mix, 2 μL of 27F/1492R primer (1 mmol/L), and 20 μL of ultrapure water. The thermocycling conditions were 94 °C for 5 min; 30 cycles of 94 °C for 30 s, 58 °C for 30 s and 72 °C for 30 min; and a final extension at 72 °C for 5 min. The PCR products were separated electrophoretically in 1.5% agarose gel electrophoresis and sequenced. Then, the phylogenetic tree was constructed by the neighbor-joining method test in MEGA 7.0 version, and the bootstrap test was replicated 500 times.

### 2.6. Extracellular Enzyme Activity Assays

Protease enzyme medium (skim milk 10 g/L and agar 18 g/L), glucanase enzyme medium (glucan 5 g/L, NaNO_3_ 2 g/L, K_2_HPO_4_ 1 g/L, KCl 0.5 g/L, MgSO_4_ 1 g/L, FeSO_4_ 0.01 g/L, agar 2 g/L, 0.005% Congo red) and cellulase enzyme medium (CMC-Na 10 g/L, (NH_4_)_2_SO_4_ 4 g/L, KH_2_PO_4_ 1g/L, MgSO_4_ 0.5 g/L, CaCl_2_ 0.02 g/L, peptone 1 g/L, yeast extract 1 g/L, agar 20 g/L, 0.005% Congo red) were used as previously reported [37,38,39]. Each antagonistic bacteria strain was grown in LB broth at 37 °C until OD_600_ 1.0, then 20 μL bacterial suspension was dropped onto each medium. The plates were incubated at 37 °C for 7 d, and the diameter of any transparent circles was measured.

### 2.7. Lipopeptides Detection

The matrix *trans*-2-[3-(4-tert-butylphenyl)-2-methyl-2-propenylidene]malononitrile was dissolved at a final concentration of 30 mg/mL in 90% *v*/*v* aqueous acetonitrile solution with 0.1% formic acid. A loop of the bacterial strain was smeared onto the stainless steel 384 target plate and 2 μL matrix solution was applied on top of the sample. The samples were subjected to matrix-assisted laser desorption/ionization-time of flight mass spectrometry (MALDI-ToF MS) analysis on a Bruker Autoflex Speed mass spectrometer in reflector/positive mode. Each sample was ablated 1000 times within 1 s with a laser energy output of 70%. A mass-to-charge (*m*/*z*) window of 500–3000 was applied to monitor the production of lipopeptides. All mass data were analyzed using the FlexAnalysis software (V 3.4, Bruker Daltonik GmbH, Bremen, Germany).

### 2.8. The Expression Profile Analysis of Cotton Resistance Genes

Cotton seeds (*Gossypium hirsutum* L., cv. Coker 312) were soaked in 75% *v*/*v* ethanol for 5 min and washed with sterile distilled water twice, then sowed in sterilized soil (peat compost: vermiculite, 1:1, *w*/*w*). They were cultured in a greenhouse (28 ± 2 °C, 75% relative humidity, 16 h light/8 h dark). The antagonistic bacteria strain was shake-cultured in LB broth at 37 °C overnight and recultured (1:100) until OD_600_ 1.0. They were collected and regulated the final OD_600_ 1.8–2.0 with sterile distilled water. The roots of two true leaves seedlings washed and soaked in bacteria suspension liquid for 30 min were kept in a wet environment for 24 h. Then, the RNA was isolated from cotton roots with PLANTeasy RNA extraction kit (YPHBio, Tianjin, China) and cDNA was synthesized following the manufacturer’s instructions (TransGen, Beijing, China). The primers of cotton resistant genes using in the quantitative real-time PCR (qRT-PCR) was designed by Primer Premier (version 5): phenylalanine ammonia lyase (PAL), TGGTGGCTGAGTTTAGGAAA/TGAGTGAGGCAATGTGTGA; MAPK TTACAATCTTATTCCACACACGC/TCCCTATTTATAGAAAACCTCCC; pathogenesis-related gene (PR10) ATGATTGAAGGTCGGCCTTTAGGG/CAGCTGCCACAAACTGGTTCTCAT. Cotton small subunit ribosomal RNA gene (SSU) was selected as a standard plant control with primer sets (AACTTAAAGGAATTGACGGAAG and GCATCACAGACCTGTTATTGCC). The qRT-PCR was performed by 7500 Real Time PCR System (ABI, Waltham, MA, USA) using TransStart Top Green qPCR Supermix (TransGen Biotech, Beijing, China). The 2 ^−∆∆Ct^ method was used for relative quantification [40].

### 2.9. Disease Index

The cotton seedlings inoculated with *V. dahliae* were replanted into new pots as described previously [41]. Symptoms on seedlings at 14 days post-inoculation (dpi) were assessed as described by [42]: 0: healthy leaves; level 1: the cotyledon displaying chlorosis or wilt; 2: one or two true leaves displaying chlorosis or wilt; 3: more than three true leaves displaying chlorosis or wilt; 4: plant death or defoliation. Then, the disease index was calculated following [Ʃ (Disease scores × Number of plants with disease score)/(Total number scored × 4)] × 100 [43]. Seedling stems were slit longitudinally between the cotyledons to check for vascular discoloration using a stereoscope (Olympus MVX10, Tokyo, Japan) [44].

### 2.10. Fungal Biomass

The fungal biomass in cotton roots was quantified as previously described [43] with a minor modification. At 14 dpi, the roots were sufficiently smashed to isolate DNA by the Plant Genomic DNA Kit (TIANGEN, Beijing, China). *Verticillium* ITS1 and ITS2 regions of the ribosomal RNA genes (Z29511) were amplified by qRT-PCR using primer sets (CCGCCGGTCCATCAGTCTCTCTGTTTATAC and CGCCTGCGGGACTCCGATGCGAGCTGTAAC) and methods described previously [45].

### 2.11. Statistical Analysis

Data from 3 independent experiments were statistically analyzed with SPSS Statistics 24.0 software (SPSS, Chicago, IL, USA). Analysis of variance was used for Duncan’s multiple range test at the 5% (*p* < 0.05) level of significance between treatments.

## 3. Results

### 3.1. Isolation and Phylogenetic Relationship Analysis of Antagonistic Bacteria against V. dahliae

Of 2000 bacterial colonies from the suspensions of cotton rhizosphere soil, 88 strains exhibited strong antagonistic activity against *V*. *dahliae* on PDA. Phylogenetic analysis of the 16S rRNA sequence revealed that these strains clustered into five major clades, as indicated by different colors (Figure 1). The sequences were aligned on NCBI-Genbank databases and correspondence with *Bacillus*, *Paenibacillus Ash*, *Pseudomonas*, *Enterobacter Hornaeche and Edwards* and *Lysinibacillus*. Their ratios in total are 94.31%, 1.14%, 2.27%, 1.14% and 1.14%, respectively.

### 3.2. In Vitro Antagonism Test

Among the 88 antagonistic bacteria, the colony morphology and inhibition efficacy of top 9 strains were observed (Figure 2). According to the 16S rRNA sequences and colony morphology assay, ABLF-8 belonged to *Pseudomonas*, ABLF-18, ABLF-50, ABLF-52, ABLF-57, ABLF-86, ABLF-107 and ABLF-117 belonged to *Bacillus*, while ABLF-90 belonged to *Paenibacillus Ash* (Figure 2a). The colonies of ABLF-8 and ABLF-90 were nearly round, nontransparent with a smooth surface and regular edges, while others were not. The colony diameter of ABLF-107 was obviously largest (Figure 2b). In the *in vitro* antagonistic assay of the strains against fungal strain V991, colony diameter and mycelial development was significantly inhibited by all antagonists compared with that of the V991 control culture alone on PDA. All mycelium co-cultured with an antagonist were chaos and degraded, especially in the case of ABLF-50, which caused the greatest inhibition (Figure 2c). Thus, the results show that these antagonistic bacteria have competence to strongly inhibit the *V*. *dahliae* on culture medium.

In the pot experiments to assess efficacy, cotton seedlings were inoculated with candidate strain. Two weeks after being inoculated with V991, the plants varied in their disease index. ABLF-8, ABLF-52, ABLF-86 and ABLF-107 only affected the growth of *V. dahliae* on medium, while ABLF-18, ABLF-50 and ABLF-90 significantly inhibited the disease index (Appendix A). We further investigated the biocontrol effect of these three antagonistic strains (Figure 3). As expected, the seedlings inoculated with ABLF-18, ABLF-50 and ABLF-90 exhibited significant resistance against *V. dahliae* (Figure 3a). The disease index was approximately decreased by nearly 50% (Figure 3b). The fungal biomass of cotton roots quantified by qRT-PCR demonstrated that the fungal DNA of *V. dahliae* in antagonistic strain treated groups was almost two-fold relative to the V991 treated group (Figure 3c). The results reveal that ABLF-18, ABLF-50 and ABLF-90 strains have potential for further development to improve the plant resistance against *V*. *dahliae*.

### 3.3. Extracellular Enzyme Activity

In order to further explore the antagonistic mechanism, we detected the fungal cell wall-degrading enzyme activity produced by antagonistic bacteria on different medium (Figure 4). The antagonistic strains were inoculated on different assay plates at 37 °C for one week, respectively. The transparent circle around all the bacterial colonies were measured, demonstrating the extracellular enzyme secreted by antagonistic bacteria to degrade the fungal cell wall. The enzyme activities (cellulase, glucanase, and protease) were disparate and the ABLF-50 were distinctly lower than others. These results suggest that the fungal cell wall-degrading enzymes secreted by antagonistic bacteria contributes to suppressing the *V. dahliae*.

### 3.4. Lipopeptides Production

Bacteria of genus *Bacillus* and *Paenibacillus* were known to produce a variety of lipopeptide secondary metabolites as antifungal or plant systemic resistance inducing agents [46,47]. To investigate lipopeptide production in our antagonistic *Bacillus* and *Paenibacillus* spp., strains ABLF-18, ABLF-50, and ABLF-90 were subjected to matrix-assisted laser desorption/ionization time of flight mass spectrometry (MALDI-ToF MS) analysis. The MALDI-ToF MS spectra of *Bacillus* sp. ABLF-18 and ABLF-50 exhibited the same set of molecular ions at *m*/*z* 1047, 1061, 1075, while *Paenibacillus* sp. ABLF-90 displayed a different set of ions at *m*/*z* 1082, 1096, and 1110 (Figure 5). Individual ions within both sets have a *m*/*z* difference of 14 as compared to their adjacent ions (Figure 5), corresponding to the gain/loss of a CH_2_ functionality commonly seen among lipopeptide homologues. Structure searches of the Natural Products Atlas [48] and Norine databases [49] suggested that molecular ions at *m*/*z* 1047, 1061, 1075 correspond to the [M+K]^+^ adducts of surfactin A, B, and C, respectively, while molecular ions at *m*/*z* 1082, 1096, and 1110 are the [M + K]^+^ adducts of iturin homologues. These results indicated that production of antimicrobial lipopeptides may contribute to the antagonism effect of the strains of *Bacillus* and *Paenibacillus* spp. against *V*. *dahliae*.

### 3.5. Cotton Resistant Genes Expression

*PAL*, *MAPK* and *PR10* are critical systemic resistant genes in cotton. After 24 h inoculated with antagonistic bacteria, the RNA of cotton seedling leaves was extracted and the qRT-PCR were employed to investigate their expression levels (Figure 6). The expression levels of *PAL*, *MAPK* and *PR10* were distinctly upregulated in ABLF-18, ABLF-50 and ABLF-90, compared with the CK. *PAL* displayed the most obvious increase in expression, with a 5-fold increase in ABLF-50. *MAPK* and *PR10* also exhibited evident increase in ABLF-90, 9- and 7-fold in expression, respectively. These results demonstrated that the antagonistic bacteria could enhance the expression of cotton genes relevant to the systemic resistance.

## 4. Discussion

The rhizosphere, firstly described by Hiltner in 1904, is the closely associated soil around living plant roots and provides nutrients and energy for rhizosphere microorganisms. The number of microorganisms is generally dozens of fold higher in the rhizosphere than in the rest of the soil [50,51]. Plant-associated microbes can influence the rhizosphere microenvironment and plant growth and can be beneficial, deleterious or neutral to plant growth [52]. Beneficial bacteria make up merely 1% of the total rhizosphere bacteria, dominated by *Bacillus* and *Pseudomonas* spp. [53]. Screening rhizosphere microorganisms is indispensable in finding bacteria that can inhibit the plant pathogenic fungi and stimulate plant defense [30].

*Bacillus* is a common gram-positive bacterium, which is an important resource for the development of microbiological pesticides. At present, a series of microbiological fungicides with *Bacillus subtilis* as the main active ingredient have been developed at home and abroad and have been widely studied and applied in the biological control of plant diseases [35,54,55]. In our screening of 2000 bacterial colonies isolated from the soil from rhizosphere of the healthy cotton seedlings in the artificial *Verticillium* wilt nursery, 88 strains inhibited *V. dahliae* growth in vitro. The ratio of antagonists among the total strains was thus approximately 4%, and *Bacillus* and *Pseudomonas* spp. made up the largest proportion of the antagonists. It may be easier to extract the potential strains from the cotton rhizosphere soil in the special environment and consistent with screening tests in previous.

Nine strains were subsequently selected among of the 88 antagonistic bacteria according to the colony morphology and inhibition efficacy [56,57,58]. The colony diameter of *V. dahliae* was significantly reduced, and the growth and development of mycelium on the border was severely impaired. Although these candidate antagonistic bacteria could effectively inhibit the *V. dahliae* growth on culture medium, the antagonistic effects should be further determined by the pot experiments and are related to survival rate, propagation speed, and the colonization so on [59,60,61]. In the present study, our results demonstrated that ABLF-18, ABLF-50 and ABLF-90 were able to distinctly reduce *V. dahliae* induced disease index and fungal biomass, which belong to *Bacillus* and *Paenibacillus Ash*. To date, many studies involving in extracellular enzyme, induced systemic resistance and secondary metabolites were reported to reveal the molecular mechanism of biocontrol agents.

The fungal cell wall-degrading enzyme activity was further investigated to clarify the antagonistic mechanism. The fungal cell wall primarily consisted of cellulase, protein and other glucan structural components, which maintain the plasticity and integrity and are essential for the fungal cell shape and normal growth [62]. Moreover, the wall also contributes to fungal attachment to the host and protection against host defense responses and is thus crucial for pathogenicity [63,64]. Degradation of fungal cell walls by bacterial enzymes disrupts cell structure and degradative products, such as chitin oligomers, β-glucans and their fragments, are important mediators of pathogen/microbe-associated molecular patterns (PAMP) that are recognized by plant cell receptors to trigger the plant immunity system [65,66,67]. Chitinase, β-1,3-glucosidase, cellulase and protease, vital components of extracellular fungal cell-wall degrading enzymes, were secreted by four *Streptomyces* isolates when the sole carbon source was cell walls of *V. dahliae* [34]. The excellent biocontrol agent, *Bacillus subtilis* NCD-2 produces extracellular protease and cellulase and inhibits the fungal growth in vitro [37]. Four endophytic bacteria that are isolated from tomato plants and have evident antifungal capability, secreted protease, chitinase and other compounds [33]. In this study, all 9 bacterial isolates could produce cellulase, glucanase, and protease to degrade the fungal cell wall under in-vitro conditions, although the enzyme activities were varied among them. Furthermore, degradative products are likely recognized by plant acceptors to activate PAMP-triggered immunity (PTI) and elevate the resistance. Because cell wall structure is similar among pathogenic fungi, antagonistic strains might be capable of inhibiting a broad range of soil-borne fungi.

The bioactive secondary metabolites produced by *Bacillus* and *Paenibacillus* species, including lipopeptides, such as fengycins, surfactins and iturins, are known for their antimicrobial activity, inducing plant systemic resistance, and biofilm formations [46,68]. The structural diversity of lipopeptides arise from variations in the number, order, and identity of the amino acids and the length and shape of the fatty acid side chains. Although structurally divergent, lipopeptides are biosynthesized via the nonribosomal pathway, in which nonribosomal peptide synthetase catalyzes the assembly of the hydrophilic peptide backbone and the incorporation of the hydrophobic lipid tail [69]. We have demonstrated that *Bacillus* sp. ABLF-18, ABLF-50 and *Paenibacillus* sp. ABLF-90 were able to produce surfactin and iturin family lipopeptides during co-cultivation with *V*. *dahliae*, and the antagonistic effects of these three strains may be partially resulted from the antifungal activity of these nonribosomal peptides.

Inducing plant systemic resistance is vital to protect from fungal infection and detriment [70,71]. The host tolerance to the pathogen is enhanced, and the expression of defense genes such as *PAL*, peroxidase (*POD*), and *PR10* is induced [72,73]. *Trichoderma* can distinctly stimulate the activity of PAL, SOD and POD in the leaves against infection [74]. The expression levels of cotton *PAL*, *PPO* and *POD* were obviously elicited by the endophytic bacterium *Bacillus cereus* YUPP-10 [32]. Increasing evidence shows that this strain produce a novel antibiotic antimicrobial protein CGTase to upregulate the pathogenesis-related gene expression in cotton and transgenic *Arabidopsis thaliana*, including *PAL*, *HSR203J* and *Jaz* so on [75]. In this study, ABLF-50 triggered the abundant *PAL* expression, which indicated it activates the salicylic acid signal pathway to improve disease resistance [76]. The expression of *MAPK* and *PR10* were highest induced by ABLF-18, which suggest the Mitogen-activated protein kinases (MAPKs) signal pathway and resistance ability is obviously boosted [77,78]. Although these genes are differentially expressed, respectively, their expression levels are significantly improved by the antagonistic bacteria. Later, whether or not they can induce other additional defenses such as flavonoid biosynthesis, a reactive oxygen species (ROS) burst and phenylpropanoid biosynthesis, and are involved in plant disease resistance, will be further explored.

## 5. Conclusions

In this study, 88 antagonistic strains, isolated from the cotton rhizosphere soil in the artificially infested nursery, were identified as *Bacillus*, *Paenibacillus Ash*, *Pseudomonas*, *Enterobacter Hornaeche and Edwards* and *Lysinibacillus*. The ABLF-18, ABLF-50 and ABLF-90 strains screened from 9 strains with strong antagonistic effects on medium could evidently reduce the disease index and fungal biomass, whereafter the antimicrobial mechanism was intensively investigated. They could secrete fungal cell wall-degrading enzymes and nonribosomal peptides to inhibit the colony diameter and mycelial development, and degrade the fungal mycelium. Furthermore, they could improve the expression levels of cotton seedlings related to the resistant genes against *V*. *dahliae*. In summary, the present results nicely demonstrate that these strains could increase plant resistance, presenting enormous potential for controlling this fungus in the field.

## Figures and Tables

**Figure 1 cells-10-03588-f001:**
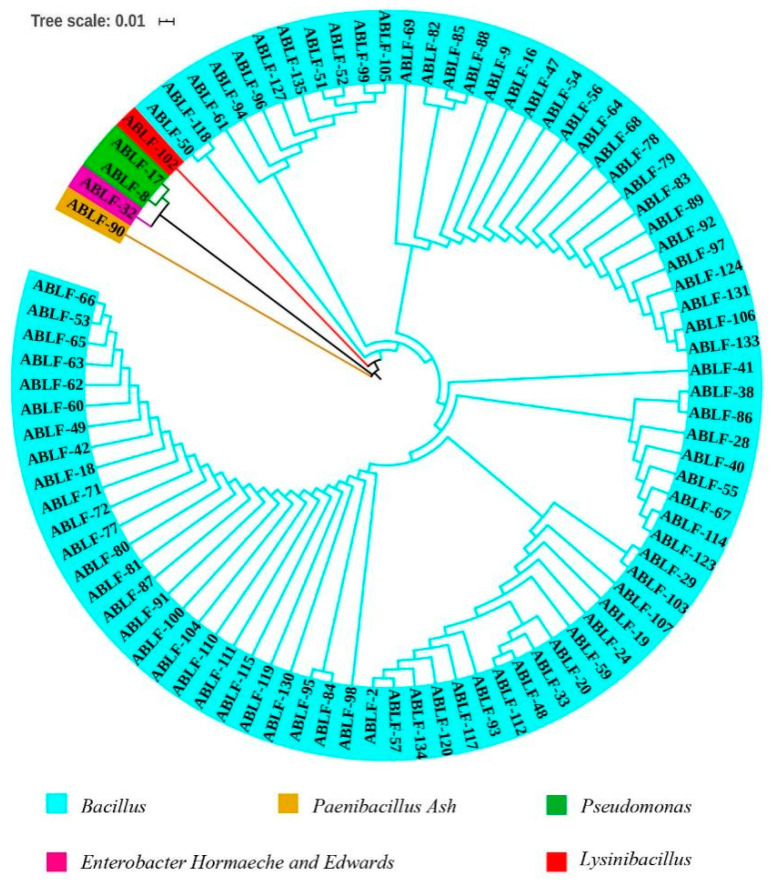
Phylogenetic analysis of candidate antagonistic bacteria against *V. dahliae*. The 16S rRNA sequences were aligned by ClustalW and phylogenetic tree was constructed with MEGA software (version 7.0, Temple University, Philadelphia, PA, USA) by the neighbor-joining method for 500 bootstraps. Bar 0.01 substitutions per nucleotide position.

**Figure 2 cells-10-03588-f002:**
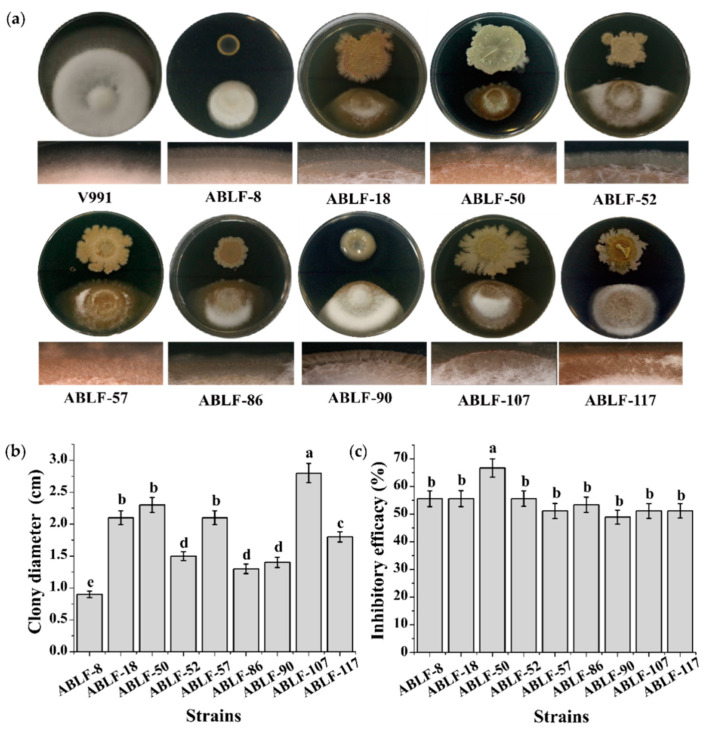
Antagonistic effect assessment of top 9 bacteria against *V. dahliae*. (**a**) Antagonistic situation. (**b**) The colony diameter. (**c**) Inhibitory efficacy. Different letters indicate significant differences between treatments according to Duncan’s test (*p* < 0.05).

**Figure 3 cells-10-03588-f003:**
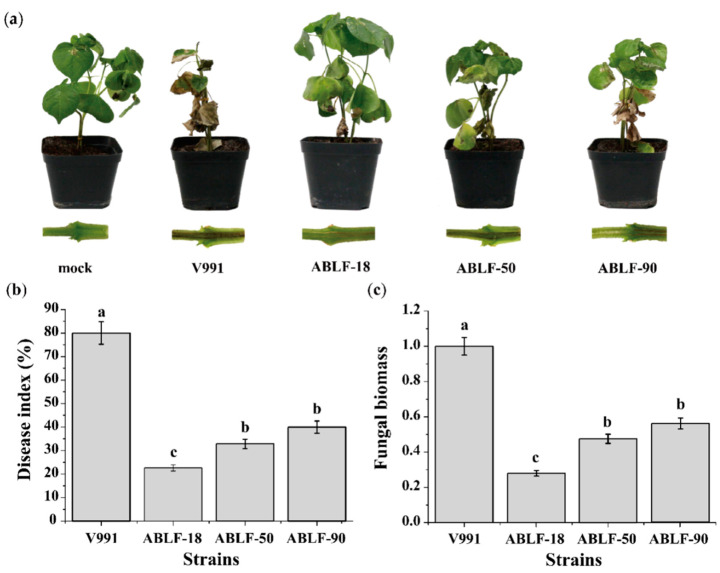
The potential biocontrol effect (**a**). Phenotype analysis of cotton seedlings. After 24 h inoculated with sterile distilled water and candidated bacterial solution, the cotton seedlings were inoculated with V991 conidial suspension and treated at 14 dpi. (**b**). The disease index analysis. (**c**). Fungal biomass detection. The cotton small subunit ribosomal RNA gene (SSU) was used as an endogenous control, while *Verticillium* ITS1 and ITS2 regions of the ribosomal RNA genes (Z29511) was selected to quantify the colonization of *V. dahliae*. Error bars represent the standard deviations from three independent experiments and different letters indicate significant differences at *p* < 0.05 as determined using a Duncan’s test.

**Figure 4 cells-10-03588-f004:**
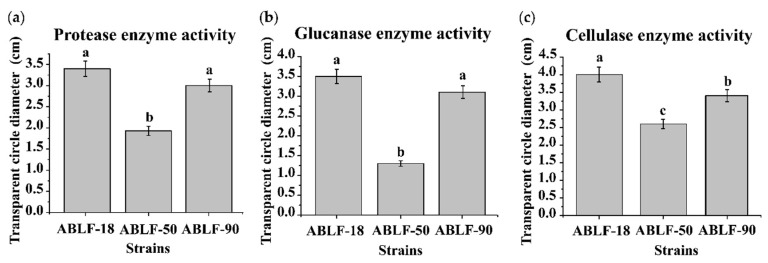
Extracellular enzyme activity analysis. The protease enzyme activity (**a**), glucanase enzyme activity (**b**) and cellulase enzyme activity (**c**) were tested on different medium. Different letters represent significant differences at *p* < 0.05 by Duncan’s test.

**Figure 5 cells-10-03588-f005:**
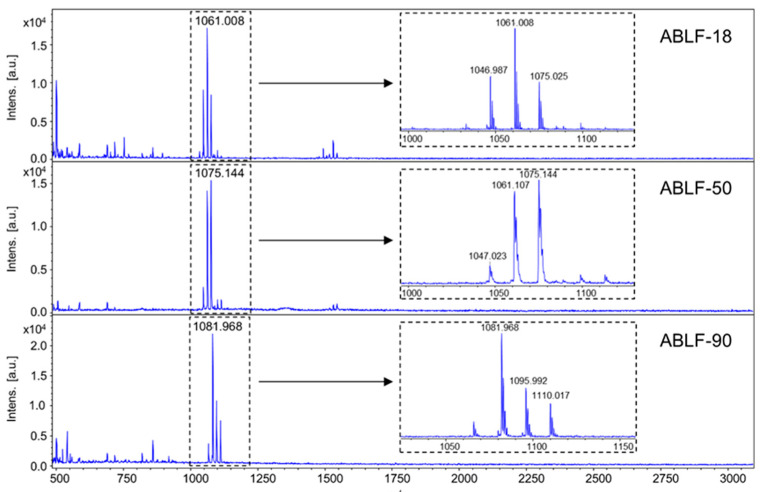
MALDI-ToF MS spectra of *Bacillus* sp. ABLF-18, *Bacillus* sp. ABLF-50, and *Paenibacillus* sp. ABLF-90.

**Figure 6 cells-10-03588-f006:**
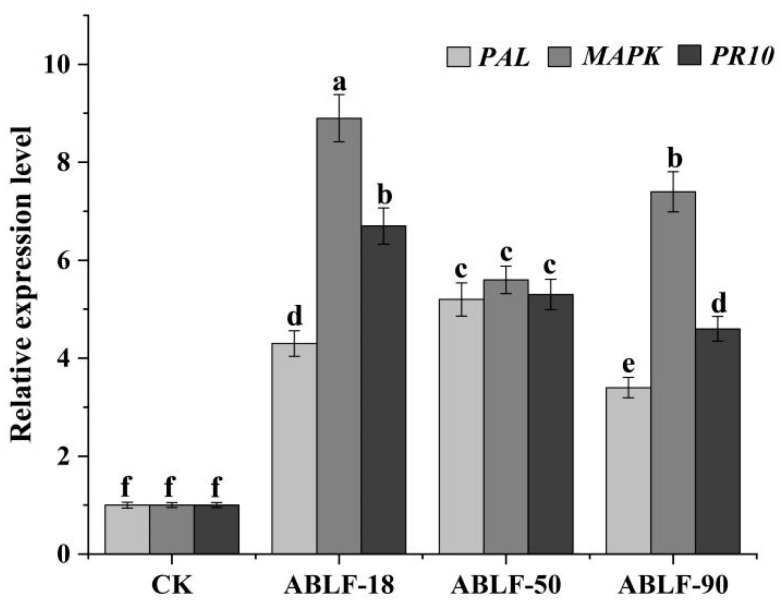
The expression patterns of cotton resistant genes. *PAL*, *MAPK* and *PR10* expression levels were examined by qRT-PCR. The RNA was extracted from the cotton roots infected with *V. dahliae* at 24 h post inoculation. Error bars represent the standard deviations from three independent experiments and different letters indicate significant differences at *p* < 0.05 as determined using a Duncan’s test.

## Data Availability

The datasets presented in this study are available upon request from the corresponding author.

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
