# Peer review of "Potential Antagonistic Bacteria against Verticillium dahliae Isolated from Artificially Infested Nursery"

_cells, 2021, doi:10.3390/cells10123588_

Round 1

Reviewer 1 Report

In recent last, microbial antagonistic have been frequently employed for the effective control of phytopathogen growth. The Present article "Potential antagonistic bacteria isolated from the artificial disease nursery as inhibitors of Verticillium dahliae" is a small piece of work and cover very limited aspect. But the experiment is well designed and clearly executed.

Author Response

Dear Reviewer,

It is in response to your suggestions regarding our manuscript entitled “Potential antagonistic bacteria isolated from the artificial disease nursery as inhibitors of Verticillium dahliae” (cells-1464179). We would like to thank you on behalf of all the authors for reviewing our manuscript. We have responded to all the suggested revisions point-by-point as below.

Responses to your comments:

In recent last, microbial antagonistic have been frequently employed for the effective control of phytopathogen growth. The Present article "Potential antagonistic bacteria isolated from the artificial disease nursery as inhibitors of Verticillium dahliae" is a small piece of work and cover very limited aspect. But the experiment is well designed and clearly executed.

Answer: Thanks for your encouraging words.

Reviewer 2 Report

Dear editor and colleagues

I have carefully read the submitted manuscript to cells, “Potential antagonistic bacteria isolated from the artificial disease nursery as inhibitors of Verticillium dahliae”

It is a manuscript describing the isolation and characterization of anti-fungal (Verticilium dahliae) bacteria and their in vitro/in vivo evaluation.

Although it is an interesting and well-designed study, still there are -quite a few- similar and published studies. As a result, the current work lacks novelty.

Moreover, according to my opinion, the submitted paper has little relevance to the aims and scopes of the cells journal (Plant, Algae and Fungi Cell Biology). In order to gain a plant biology perspective, more analytical techniques (transcriptomics/metabolomics) are in need to show how the application of microorganism interacts with the homeostasis of cotton plants. Based on the current analyses the work is largely descriptive.

As a result, my suggestion is that the manuscript should be rejected and resubmitted to a more suitable journal

Author Response

Dear Reviewer,

It is in response to your suggestions regarding our manuscript entitled “Potential antagonistic bacteria isolated from the artificial disease nursery as inhibitors of Verticillium dahliae” (cells-1464179). We would like to thank you on behalf of all the authors for reviewing our manuscript. We have responded to all the suggested revisions point-by-point as below.

Responses to your comments:

Although it is an interesting and well-designed study, still there are -quite a few- similar and published studies. As a result, the current work lacks novelty.

Moreover, according to my opinion, the submitted paper has little relevance to the aims and scopes of the cells journal (Plant, Algae and Fungi Cell Biology). In order to gain a plant biology perspective, more analytical techniques (transcriptomics/metabolomics) are in need to show how the application of microorganism interacts with the homeostasis of cotton plants. Based on the current analyses the work is largely descriptive.

As a result, my suggestion is that the manuscript should be rejected and resubmitted to a more suitable journal.

Answer: Thanks for your kind advice.

We carefully read the aims and scope of cells and completed our manuscript according the Instructions for Authors. In our study, firstly, the antagonistic bacteria against Verticillium dahliae were isolated from the artificially infested nursery by the inhibitory effect of pathogenic bacteria on medium; secondly, three strains could significantly increase the resistant ability against V. dahliae; finally, the molecular mechanism were further investigated. Taken together, our study is involved in plant and fungi cell biology and we consider it is suit to submit this journal.

Reviewer 3 Report

Dear Authors,

The manuscript 10.3390 “Potential antagonistic bacteria isolated from the artificial dis-2 ease nursery as inhibitors of Verticillium dahliae” presents interesting data on the application of antagonistic bacteria as biocontrol agent against Verticillium dahliae.

If I correctly read the paper, 88 strains mainly belonging to the genera Bacillus strongly suppressed the colony diameter of V. dahliae up to 50%. Compared to the control, ABLF-18, ABLF-50 and ABLF-90 reduced up to 40-70% the disease index of cotton plants inoculated with V. dahliae. The proposed antagonists secrete extracellular enzymes, including cellulase, glucanase, and protease, and lipopeptides, such as surfactin and iturin, as biocontrol mechanism. Moreover, candidate strains elicited PAL, MAPK and PR10 in inoculated cotton plants.

My general impression of this study is positive, but its presentation in the form of a manuscript requires adjustments.

At present, the English language in the text is not uniform. Some parts are well written and flow clearly; other parts do not flow so clearly. Several misspelling are also present.

Arrange the manuscript (reference indications) following the journal's instructions.

Use the International System of Units.

The following suggestions could improve the manuscript.

The terms "artificial disease nursery" are not clear, I suggest "artificially infested nursery"

I suggest the title:

Potential antagonistic bacteria against Verticillium dahliae isolated from artificially disease nursery

Abstract

Reduce the general information on bacterial resources database.

Is "Cohn" referred as the specific epithet or as the Author?

Keywords

I suggest the following:

Extracellular enzyme; nonribosomal peptides; cotton; resistant gene expression; biological fungicides

Introduction

Explain the interactions between V. dahliae and cotton plants.

Materials and Methods

Lines 114-115: Use “the previous methods [32] with minor modifications.” instead of “the previous methods with 114 minor modifications [32].”

Line 117: Use “incubated at” instead of “cultured”.

Line 120: What does "The V. dahliae spores was" mean?

Line 121: Use “adjusted to” instead of “regulated into”.

Line 124: Use “further” instead of “in-depth”.

Line 126: Insert a reference or describe the DNA of antagonistic bacteria extraction procedure.

Line 136: Use “Extracellular enzyme activity assays” or “Extracellular enzyme activity” instead of “Enzyme activity assays of extracellular enzyme”.

Lines 137-144: I suggest:

Protease assay medium (skim milk 10 g/L and agar 18 g/L), glucanase assay medium (glucan 5 g/L, NaNO3 2 g/L, K2HPO4 1 g/L, KCl 0.5 g/L, MgSO4 1 g/L, FeSO4 0. 01 g/L, agar 139 2 g/L, 0.005% Congo red) and cellulase assay medium (CMC-Na 10 g/L, 140 (NH4)2SO4 4 g/L, KH2PO4 1g/L, MgSO4 0.5 g/L, CaCl2 0.02 g/L, peptone 1 g/L, yeast extract 141 1 g/L, agar 20 g/L, 0.005% Congo red) were used as previously reported [34-36]. Each antagonistic bacteria strain was grown in liquid LB medium at 37 °C until OD600 1.0.

Lines 144-145: What does "20 μL of bacterial liquid culture was injected into 5-mm plugs taken from the different assay medium" mean?

Line 147: Use “Lipopeptides detection” instead of “Detection of lipopeptides”.

Line 180: delete “analysis”.

Line 190: delete “analysis”.

Results

This section is particularly difficult to read.

Re-arrange all the figure legend. Figures must be clear and easy to understand without extra explanation.

If it is possible, use an unique Relative expression level (11) for each patterns of gene.n particular

Line 203: delete “, screening  against V. 203 dahliae”.

Line 218: Use “In vitro antagonism test” instead of “Antagonism evaluation and inhibition assay”.

Line 235: delete.

Line 259: Use “Extracellular enzyme activity” instead of “Extracellular enzyme activity secreted by antagonistic bacteria”.

Line: 273: Use “Lipopeptides production” instead of “Lipopeptides produced by the antagonistic bacteria”.

Line 293: Use “Cotton resistant genes expression” instead of “Antagonistic bacteria could facilitate the expression levels of cotton resistant genes”.

Author Response

Dear Reviewer,

It is in response to your suggestions regarding our manuscript entitled “Potential antagonistic bacteria isolated from the artificial disease nursery as inhibitors of Verticillium dahliae” (cells-1464179). We would like to thank you on behalf of all the authors for reviewing our manuscript. We have responded to all the suggested revisions point-by-point as below.

Responses to your comments:

  1. At present, the English language in the text is not uniform. Some parts are well written and flow clearly; other parts do not flow so clearly. Several misspelling are also present.

Answer: Thanks for your suggestion. We carefully revised the manuscript and did our best to improve it. Extensive English editing has now been completed by Beth E. Hazen (PhD), a professional scientific editor and native English speaker.

  1. Arrange the manuscript (reference indications) following the journal's instructions.

Answer: We have carefully followed the journal’s instructions and revised the whole manuscript.

3.Use the International System of Units.

Answer: We and Dr. Hazen have carefully checked all units and revised them as needed.

  1. The terms "artificial disease nursery" are not clear, I suggest "artificially infested nursery"

Answer: We changed it in the whole manuscript as per your instruction.

  1. I suggest the title: Potential antagonistic bacteria against Verticillium dahliaeisolated from artificially disease nursery.

Answer: It is a good idea. The title has been changed.

  1. Reduce the general information on bacterial resources database.

Answer: Done.

  1. Is "Cohn" referred as the specific epithet or as the Author?

Answer: We apologize for the mistake; “Cohn” is the author of the genus.

  1. I suggest keywords as the following: Extracellular enzyme; nonribosomal peptides; cotton; resistant gene expression; biological fungicides.

Answer: We have used the suggested keywords.

  1. Explain the interactions between V. dahliaeand cotton plants in introduction.

Answer: It is a good suggestion to connect the V. dahliae and cotton plants and better understand this study. We added the interaction between them in the introduction. Please see line 52-54.

  1. Lines 114-115: Use “the previous methods [32] with minor modifications.” instead of “the previous methods with minor modifications [32].”

Answer: We have moved it. Please see line 138-139.

  1. Line 117: Use “incubated at” instead of “cultured”.

Answer: We have changed it. Please see line 142-143.

  1. Line 120: What does "The V. dahliae spores was" mean?

Answer: We have corrected it: “V. dahliae spores were”. Please see line 145.

  1. Line 121: Use “adjusted to” instead of “regulated into”.

Answer: As recommended, we have changed it. Please see line 146.

  1. Line 124: Use “further” instead of “in-depth”.

Answer: Done. Please see line 149.

  1. Line 126: Insert a reference or describe the DNA of antagonistic bacteria extraction procedure.

Answer: We have added a relevant reference. Please see line 151-152.

  1. Line 136: Use “Extracellular enzyme activity assays” or “Extracellular enzyme activity” instead of “Enzyme activity assays of extracellular enzyme”.

Answer: We revised as “extracellular enzymatic activity assays”. Please see line 163.

  1. Lines 137-144: I suggest: Protease assay medium (skim milk 10 g/L and agar 18 g/L), glucanase assay medium (glucan 5 g/L, NaNO32 g/L, K2HPO41 g/L, KCl 0.5 g/L, MgSO4 1 g/L, FeSO4 0. 01 g/L, agar 2 g/L, 0.005% Congo red) and cellulase assay medium (CMC-Na 10 g/L, (NH4)2SO4 4 g/L, KH2PO4 1g/L, MgSO4 0.5 g/L, CaCl2 0.02 g/L, peptone 1 g/L, yeast extract 1 g/L, agar 20 g/L, 0.005% Congo red) were used as previously reported [34-36]. Each antagonistic bacteria strain was grown in liquid LB medium at 37 °C until OD600 1.0.

Answer: We have revised the text mostly as suggested and as edited by Dr. Hazen, except for “LB broth” was used. Please see line 164-172.

  1. Lines 144-145: What does "20 μL of bacterial liquid culture was injected into 5-mm plugs taken from the different assay medium" mean?

Answer: I am very sorry for that the expression is not clear. We have revised. Please see line 172-173.

  1. Line 147: Use “Lipopeptides detection” instead of “Detection of lipopeptides”.

Answer: we have changed it. Please see line 176.

  1. Line 180: delete “analysis”.

Answer: Thank you for pointing it out, we have changed it. Please see line 212.

  1. Line 190: delete “analysis”.

Answer: Agreed, we have deleted it. Please see line 224.

  1. This section is particularly difficult to read. Re-arrange all the figure legend. Figures must be clear and easy to understand without extra explanation. If it is possible, use an unique Relative expression level (11) for each patterns of gene.n particular.

Answer: Thanks for your kind advice. We have re-arranged all figure legends and used unique patterns to show the relative expression of each gene. Please see figure 6.

  1. Line 203: delete “, screening  against V. dahliae”.

Answer: Done. Please see line 240-241.

  1. Line 218: Use “In vitroantagonism test” instead of “Antagonism evaluation and inhibition assay”.

Answer: We have changed it. Please see line 257.

  1. Line 235: delete.

Answer: Done. Please see line 277.

  1. Line 259: Use “Extracellular enzyme activity” instead of “Extracellular enzyme activity secreted by antagonistic bacteria”.

Answer: We have changed it as described earlier. Please see line 304.

  1. Line: 273: Use “Lipopeptides production” instead of “Lipopeptides produced by the antagonistic bacteria”.

Answer: Our editor changed it to “lipopeptide production”. Please see line 318.

28.Line 293: Use “Cotton resistant genes expression” instead of “Antagonistic bacteria could facilitate the expression levels of cotton resistant genes”.

Answer: As recommended, we have changed it. Please see line 339.
